# Allogenic Use of Human Placenta-Derived Stromal Cells as a Highly Active Subtype of Mesenchymal Stromal Cells for Cell-Based Therapies

**DOI:** 10.3390/ijms22105302

**Published:** 2021-05-18

**Authors:** Raphael Gorodetsky, Wilhelm K. Aicher

**Affiliations:** 1Biotechnology and Radiobiology Laboratory, Sharett Institute of Oncology, Hadassah-Hebrew University Medical Center, Jerusalem 91120, Israel; 2Center of Medical Research, Department of Urology at UKT, Eberhard-Karls-University, 72076 Tuebingen, Germany

**Keywords:** mesenchymal stromal/stem cells (MSC), placenta-derived mesenchymal stromal cells (hPSCs/pMSCs), bone marrow MSCs (bmMSC), adipose tissue stromal cells (atMSCs), pro-regenerative effects, xenogeneic cell delivery, regenerative cell therapy, cell-based immune modulation

## Abstract

The application of mesenchymal stromal cells (MSCs) from different sources, including bone marrow (BM, bmMSCs), adipose tissue (atMSCs), and human term placenta (hPSCs) has been proposed for various clinical purposes. Accumulated evidence suggests that the activity of the different MSCs is indirect and associated with paracrine release of pro-regenerative and anti-inflammatory factors. A major limitation of bmMSCs-based treatment for autologous application is the limited yield of cells harvested from BM and the invasiveness of the procedure. Similar effects of autologous and allogeneic MSCs isolated from various other tissues were reported. The easily available fresh human placenta seems to represent a preferred source for harvesting abundant numbers of human hPSCs for allogenic use. Cells derived from the neonate tissues of the placenta (f-hPSC) can undergo extended expansion with a low risk of senescence. The low expression of HLA class I and II on f-hPSCs reduces the risk of rejection in allogeneic or xenogeneic applications in normal immunocompetent hosts. The main advantage of hPSCs-based therapies seems to lie in the secretion of a wide range of pro-regenerative and anti-inflammatory factors. This renders hPSCs as a very competent cell for therapy in humans or animal models. This review summarizes the therapeutic potential of allogeneic applications of f-hPSCs, with reference to their indirect pro-regenerative and anti-inflammatory effects and discusses clinical feasibility studies.

## 1. Development of the Concept of Cell Therapy for Pro-Regenerative Treatments

### Stem Cells: Sources and Proposed Roles in Tissue Regeneration

The definition of stem cells is not straightforward. The relevant qualities of such cells depend initially on their proliferative potential and the expectations for their trans-differentiation to mature cells of different target tissues [1]. The possibility of expanding these cells in vitro rendered them promising candidates for different potential applications in regenerative therapy. The anticipated genetic predisposition of the delivered cells is based on inherited genetic and epigenetic differentiation potential, as well as the environmental conditions in the implantation site [2].

The high interest in stem cells rose based on early experiments and relevant clinical application, based on harvesting and implanting BM hematopoietic stem cells (HSC) [3,4]. The IV injected progenitors homed to the BM to produce a wide range of hematopoietic cell phenotypes as a basis for the recovery of the whole failing hematopoietic system [4]. These findings incited the use of other stem cell sources for pro-regenerative therapy of various other failing target tissues [5,6]. Unlike the recovery of the hematopoietic system by HSC in remote IV delivery, the expectations for their homing to the target malfunction tissues to replace the damaged cells has not materialized. Rather, most of the cells were trapped immediately in the lungs’ capillaries as well as other highly vascularized tissues [7]. In contrast, the regenerative potential of other stem cells has been documented in many studies upon local administration (see below).

Regenerative cell therapies in the fast-emerging field of tissue engineering are based on the anticipation that delivered cells rebuild the affected tissues [8,9,10,11,12,13,14,15]. Various technologies tested to introduce adequate stem cells for the construction of functioning tissues and organs, both in vitro and in vivo, have had only minimal success [8,16,17,18,19,20,21,22,23]. In most cases, in vitro-constructed non-vascularized 3D tissue-engineered complex tissues based on differentiated stem cells seemed to fade upon their implantation. This failure derives from the lack of immediate vascularization and supply, which is an inherent basic obstacle that affects the survival and growth of the implanted cellular tissue constructs. This affects certain tissue types, such as cartilage, less, as the degree of vascularization is less critical for the survival of chondrocytes in 3D tissue structures [17,24,25,26,27,28]. Other directions, where cells only were introduced, with the expectation that they serve as “spare parts” for re-cellularization of ill tissues [29,30,31,32], have not turned out to be highly applicable. 

One of the major promising breakthroughs in stem cells research was the introduction of multipotent embryonic stem cells (ES cells), isolated from the gastrula at an early stage of human fetal development [2,33]. These multipotent ES cells, as well as induced stem cells (iPS cells, which present ES cell features), present similar trans-differentiation potentials [34,35]. Due to these abilities the pluripotent stem cells were initially posed as a promising basis for various directions in tissue engineering [36,37,38,39]. Up to now, ES and iPS cells have had limited clinical applications due to biosafety concerns [2,40,41,42]. The few applications that may be considered, but not approved, for clinical application include projects such as the attempt to repair genetic skin disorders [43] or to regenerate the pigment epithelium [40,41]. In addition to ethical concerns, a limitation for clinical application of ES and iPS cells can be attributed, as least in part, to the risk of uncontrolled differentiation to tumors and teratomas [44,45,46,47]. However, part of this hurdle may be overcome by novel protocols generating iPSCs without recombinant viral vectors, but their pluripotent nature may still pose a problem of a formation undesired tissues. The main current applications of different types of MSCs for therapeutic uses derive from the highly significant pro-regenerative and immune-modulatory paracrine effects [48,49,50,51,52,53,54,55]. These findings diverted gradually the focus of the field of stem cells from tissue engineering to therapies based on the paracrine effects of MSCs. In such cell therapies, the activity of the cells seemed to be less dependent on the “stemness” of the cells and their differentiation potential and was associated more with the indirect effects mediated by the secretome of the therapeutic cells [56,57,58,59,60,61]. 

## 2. Therapies Based on Mesenchymal Stem/Stromal Cell (MSC) Injections

### 2.1. Properties and Phenotype of bmMSCs and MSCs Isolated from Other Tissues

The fibroblast-like matrix-adherent colony-forming cells, referred to as mesenchymal stromal or stem cells (both under the widely used abbreviation MSCs), were initially described several decades ago. The bona fide description of bmMSCs as stem cells is based on their self-renewal capability and their ability to differentiate to cells of mesodermal tissues, mainly osteoblasts, chondrocytes, and adipocytes [45,62,63]. The main advantage in such a stable cell phenotype was that the implant of bmMSCs from different mesenchymal tissues, unlike ES and IPS cells, seemed to be less associated with risks of forming teratomas or uncontrolled malignant transformation [45,64,65,66]. 

bmMSCs and stromal cells harvested from other tissues have been tested for regenerative medicine in various animal disease models. Though the bmMSCs differentiated into several mesodermal cell phenotypes in vitro, in vivo studies reported that bmMSCs contributed to bone formation, while differentiation to other cells such as chondrogenesis remains questionable [27,67,68,69]. On the other hand, the more promising direction has been based on the accumulated findings showing that the MSCs induced a broad range of anti-inflammatory, immune-modulatory effects [70,71,72,73,74,75,76,77,78,79,80,81], and pro-regenerative effects [82,83,84,85,86], which both set a basis for more practical therapeutic applications [87,88]. 

### 2.2. Shared Features and Variations in the Phenotype of MSCs from Different Tissue Sources 

In general, all MSCs seem to share a typical set of common mesenchymal cell surface antigens, which are in part different from the markers of other cell lineages, such as hematopoietic and endothelial cells [45,89,90,91]. Based on the accumulated data in recent years, it is preferred to describe MSCs as mesenchymal stromal cells, without referring to their “stemness” as a basis for both their pro-regenerative activity [68,92,93,94] and their potential role as building blocks of the affected tissues [89,90,91]. In this context, early studies showed that injection of MSCs or dermal fibroblasts may accelerate tissue regeneration in a model of wound healing in irradiated skin [95]. This is similar to the effects so far attributed mainly to bmMSCs [96,97], which are commonly considered as the gold standard of an optimal source of active MSCs [1,53,98,99,100,101,102,103,104,105]. However, this claim is not always justified, since MSCs from different other tissues, such as PSCs, have shown equivalent or even higher potency, as described below.

### 2.3. Characterization of Different Tissues’ MSCs Based on Shared Surface Protein Markers 

Major efforts along the last few decades identified alternative tissue sources for isolating MSCs for regenerative therapy application [64], including adipose tissue [106,107], cord blood [108,109], skin [110], vascular tissues [111], the roots of shed exfoliated deciduous teeth [112,113], and term placenta [114,115,116,117,118,119,120,121,122], including the umbilical cord and Wharton’s jelly [118,123,124,125,126]. 

MSCs isolated from these tissues by standard techniques may present in a blend of different cell types, enriched by cell adhesion and the choice of expansion medium. By definition, matrix adhering cells expressing the MSCs’ markers without expression of any of the specific hematopoietic and endothelial cells’ markers are categorized as MSCs [127,128,129]. Specifically, some surface markers are considered as a proof of the identity of the isolated MSCs, including CD73, CD90, and CD105 and complemented by other cell surface markers, such as CD29, CD54, CD146, CD102, CD166, and CD271 [87,116,130] (Figure 1). The proposed interpretation of MSC markers as an indication of their differentiation potential does not seem to be adequately substantiated [131,132,133]. A meta-analysis classifying MSCs based on a database of more than 50 published studies showed discrimination of MSCs from non-MSCs with more that 95% accuracy by the above markers’ expression [134]. Additional MSCs-related markers include CD73, CD105, CD106, and CD140 (PDGF receptor B). Furthermore, CD44 and CD90 were claimed to discriminate MSCs from fibroblasts [134,135]. 

In their gene expression level, cells classified as bmMSCs expressed 425 genes at significantly different levels compared with non-MSC fibroblasts [134]. A panel of experts proposed, based on a review of bmMSCs research status, a set of consensus surface markers for bmMSC [64]. This includes CD90 and CD44, which are dominantly expressed on fibroblasts, while hPSCs presented a distinctly high expression for CD73 [135]. CD146 expression in bmMSCs was reported in MSCs in all investigated culture conditions [136], but its expression on hPSCs depends on cell culture conditions (data not shown). CD166 seemed to be a more specific marker on hPSCs, differentiating them from most other types of mesodermal stromal cells [116].

### 2.4. Characterization of MSCs from Different Sources by Their Secretion Profiles

MSCs isolated from different tissue sources seem to have similar a secretome, with variations in the amount and proportions of the different components [137,138,139]. In addition to the inherent differences between the different sources of the cells, this parameter may also depend on culture and growth conditions of the cells [59,61,88,140,141,142]. The main variations between the different types of MSCs may derive also from the sensitivity to stress signal receptors on the MSCs [143], although this point needs further investigation. Other factors may derive from the variability in the rate and efficiency of the protein synthesis machinery of the different MSCs [144]. These parameters may dictate the yield and ratio of the relevant secreted proteins per activated cell. Nevertheless, in the lack of sufficiently adequate comparative studies, this issue deserves further research. Of note is that a major factor in the effect of the delivered cells might depend on their survival in allogeneic/xenogeneic delivery to the target tissues of the treated recipient [58]. The hPSCs from the fetal tissues of the placenta may have an advantage, since they have lower HLA expression, which may explain why during long pregnancy the fetal placental tissues are not rejected by the non-matched maternal immune system, which is shared by the vascular system of the maternal placenta. The specific relevant properties of hPSCs, which may somehow differ from regular MSCs secretome, are discussed below.

Recent studies have attempted to overcome the limited ability of identifying and discriminating between different types of MSCs by using advanced molecular transcriptome and proteasome analyses [145,146,147]. Transcriptomic profiling suggested that expression of the adhesion molecule CD106 (VCAM1) and of the leptin receptor (LEPR) could serve as bmMSCs’ specific gene signatures, while the glycoprotein CD226 and the adhesion molecule CD56 (NCAM1) could serve as gene signatures of hPSCs. In contrast, no surface molecule has so far been proposed to define specifically atMSCs or MSCs from fibroblasts of skin or other tissues [135]. Comparing the transcriptome of bmMSCs with ESCs and ESC-derived clones of MSCs, open reading frames of some 870 genes were found to be significantly elevated in mesenchymal cells, specifically in MSCs [145]. Some studies compared the transcriptome of human bmMSCs with other types of MSCs, such as hPSCs [116,148]. Significant differences in some homeobox genes were shown in placental cells. Specific transcripts such as PSG1–PSG7, PSG9, and nestin were higher, while most notably, Runx2 and Twist were significantly lower in pMSC, explaining their rather low osteogenic differentiation potential in vitro [148]. Other studies focused on comparing human MSCs from different sources, including BM, AT, and placenta with structural tissue fibroblasts by transcriptomic profiling [135]. However, a consensus on these findings still needs to be established, especially since the transcriptome of the cells may depend on the degree and extent of stress signals to which the cells are exposed, as shown for instance for the profile of related protein secretion in xenogenic hPSCs injection for mitigation of acute radiation syndrome [114]. The bmMSCs were compared to hPSCs following cell expansion in cultures of high passages by the attachment to specific peptides; about 80–90% of hPSCs populations attached to a fibronectin-derived or randomly designed synthetic peptide while bmMSC populations seemed to attach less to these peptides [149]. This may relate to significant differences in the expression of integrins between bmMSCs and hPSCs [148]. 

Since most cell surface molecules investigated so far seem not unique to specific MSCs populations, they could not be used for enrichment of any specific type of MSCs from a mixture of mesenchymal cell preparations. Still, differences in cell attachment properties could be used to differentiate between MSCs from different sources. For instance, culturing hPSCs on fibrin-coated dishes yielded low cell attachment to the surface and the hPSCs stayed compact and round, while bmMSCs, under the same conditions, showed a flat, spindle-shaped appearance [78]. This conformed to earlier study on the difference between bmMSCs and hPSCs with respect to different cell–matrix interactions. Such differences may contribute to directed cell migration and may possibly induce some homing of MSCs to different niches, although this approach still needs to be substantiated [149]. 

The cloning and systematic expansion of human bmMSCs and detailed analyses showed differences in phenotype plasticity characteristics in vitro [150]. The current understanding is that the key benefit of using different types of MSCs for cell therapies is associated with their paracrine effects, as outlined above [60,69,114,151,152,153,154,155]. In addition, extracellular vesicles, i.e., microvesicles and exosomes, may carry a significant part of regenerative factors from the MSCs to neighboring cells, as well as to the circulation [156,157,158], thus opening new avenues for cell-free MSC-therapy, using supernatants of these cells collected under GMP-compliant conditions [114,159,160]. A great portion of the secreted factors of MSCs are packed in nanoparticles such as exosomes, which are typically 50–100 nm in diameter. Exosomes derived from endosomes are released following a fusion of late endosomes with the cell membrane. The proteins can also be released in micro-vesicles generated directly by budding from the cell membrane [161], which may reach sizes up to 1 μm. Therefore, extracellular vesicles may present some typical cell membrane-bound surface antigens, such as adhesion molecules integrin β1 (CD29) and glycoprotein CD44, ecto-5′-nucleotidase (CD73), or endoglin, a component of the TGF-β receptor (CD105) complex [158]. They also contain cytoplasmatic proteins, including components of intracellular signaling such a kinases or cytokines, messenger RNA, and small micro-RNA molecules (miRNA or miR). The released extracellular vesicles contain a variable panel of proteins whose composition depends on their source (BM, AT, or placenta) and activation status of the respective cell. This difference in vesicle cargo may have specific relevance to the regenerative potential of the MSCs when the paracrine factors may play a role in specific clinical situations.

Interestingly, extracellular vesicles from different sources may also have different clinical potential. The atMSCs seemed to have superior regeneration potential for angiogenesis due to secretion of IGF-1, VEGF, and TGF-β1. In vitro release of such vesicles induces the expression of angiopoetin-1 in endothelial cells. One may hypothesize that PSCs-derived micro-vesicles will contribute to angiogenesis as well [130]. A major cargo of the micro-vesicles also contains regulating miRNAs that may contribute to the effect of MSCs treatment and PSCs [88,153,162]. Here also, accumulating evidence suggests that the patterns of miRNAs may differ in MSCs isolated from different sources. For instance, miR125 was prominent in bmMSCs but miR494, which was detected in other MSCs types, was not detected [162] (Figure 1). 

### 2.5. MSCs for Modification of the Immune System Activity and Reduction of Inflammatory Responses

A major field of MSCs is dedicated to investigating their effects on the immune system in various preclinical and clinical studies. This includes graft-versus-host disease [163] and autoimmune diseases, such as lupus and arthritis and even prohibition of some cancer cell proliferation [164,165,166]. A potential benefit of MSCs treatment was also proposed for other conditions. 

In early studies it was noted that MSCs, unless specifically activated, express very few MHC class I antigens and no MHC class II molecules [167]. They were also found to express no T-lymphocyte co-stimulatory molecule CD80 (alias B7-1), which is expressed on dendritic cells presenting an antigen in the MHC context and as activating signal to T cells via CD28. The lack of MHC class I and class II molecules on MSCs, in combination with lack of CD80 and CD86 may strengthen the immune modulation of these cells by inhibiting maturation of dendritic cells and suppressing proliferation and function of T, B, and natural killer cells [167]. 

Preliminary data suggest that human MSCs express additional immune-checkpoint antigens, which further augment their anti-inflammatory action. Activation of immune cells through toll-like receptors (TLR) activates the different paths of the innate and adaptive immune responses, but activation of TLR on MSCs enhanced their immunosuppressive potential [168]. Their activity following their potential infiltration into tumors may also result in their protection from the immune system [169].

Another interesting emerging direction in cell therapy relates to the role of mitochondria in the regenerative process. It was suggested that implanted cells may not only contribute to the health of the mitochondria in the cells of the target tissue but may even transfer healthy mitochondria to the repaired failing tissues [170,171,172].

## 3. Application of hPSCs from Mature Placentae for Tissue Regeneration and Immune Modulation

Unlike the BM, the human placentae are an easily available tissue source of fully differentiated human mesenchymal cells [173]. Figure 2 shows the basic tissue structure of a full-term placenta, with special reference to the mesenchymal tissue compartment from which stromal cells could be isolated. In view of the vast availability of the disposable human full-term placentae, many studies focused on this source for isolation of MSCs [174,175,176,177,178,179].

### 3.1. The Isolation of hPSCs from Full-Term Human Placenta and Their Expansion for Pre-Clinical and Clinical Studies

The hPSCs are often termed “placental mesenchymal stem cells” (pMSCs) based on the tendency to label any mesenchymal stromal cells isolated from various other tissue sources as “MSC”. Such an assumption is based on the expectation that all MSCs share some properties and could serve as alternative for bmMSCs in cell therapies [118,120,173,180,181,182,183,184,185,186]. Nevertheless, the placenta presents as a complex tissue generated as a whole from two individuals. Cells isolated from placentae therefore come as complex blends containing probably a higher variety of cell types when compared, e.g., to blends of BM or adipose tissue-derived stromal cells [187].

Early studies expected the hPSCs to serve as building blocks of damaged tissues [182,187,188,189,190,191,192,193], but they seem to have limited differentiation potential [120,126,137,148,181,184,194,195,196]. The activity of hPSCs in allogeneic and xenogeneic administration seems to derive from their rich secretome, activated in response to a wide range of stress signals [114,130,180,197,198,199,200,201,202,203]. The hPSCs isolated from the whole placentae and further expanded tend to produce cultures primarily composed of cells from the maternal placental tissues (m-hPSCS) (Figure 1) [115,116,204]. Under careful extraction of the fetal tissues, fetal only hPSCs (f-hPSCs) can be enriched. 

A major cellular component of the placental tissues is the multinuclear syncytium of trophoblasts mixed with supporting cells. Only isolated trophoblast stem cells in early gestation could be cultured in vitro as organoids for the study of different trophoblast sub-populations. Nevertheless, it is not feasible to culture and expand trophoblasts from a mature full-term placenta [130,205].

The hPSCs isolated from the fetal versus maternal placenta and expanded separately differ in some features of their phenotype, their secretome, and activity. The shape of both types of hPSCs also differs from fibroblasts or bmMSCs (Figure 3). The hPSCs are isolated from a placenta from two interlaced organs derived from two allogeneic individuals—the maternal placental tissues with the maternal genotype and the fetal placenta tissues, where the cells originate from the tissues, with the genotype of the developing fetus. The hPSCs isolated from the two distinct tissue sources of the placenta seem to demonstrate different properties and pro-regenerative activity [206,207,208].

The identification of fetal hPSCs is not straightforward [118,122,125,208,209,210,211,212,213,214]. In the case of a male newborn, the validation of the hPSC origin can be based on gene expression or detection of Y-chromosome staining [116,215] (Figure 4A). The f-hPSCs can be isolated from different fetal placental tissues [216,217]. Some studies have examined the difference between hPSCs from the chorionic plate and the tissues fused to it, such as the cord and Wharton jelly. However, significant differences between the cells from various fetal placental tissues could not be demonstrated [83,217,218,219,220,221,222,223]. Some studies therefore proposed the use of hPSCs from Wharton Jelly, an immune-privileged tissue from which, by definition, all hPSCs are of fetal origin and apparently with similar phenotype as chorion derived f-hPSCs [199,217].

When isolating hPSCs from adjacent fetal and maternal tissues, even when the source of fetal tissues is carefully dissected from the chorionic plate, a mixture of maternal and fetal hPSCs and cells with heterogeneous properties can be anticipated. We assume that one of the subpopulations may overgrow the culture and thus reduce the proportion of the other [224]. Nevertheless, when the tissue is carefully sampled only from the chorionic plate, the cord, or the Wharton Jelly, the isolated hPSCs yield preparations enriched for f-hPSCs [116,206,219,225,226]. This may have a major impact on the properties of cells produced, as f-hPSCs are known for higher regenerative potencies than m-hPSC, at least for inducing the regeneration of the depleted BM. But careful separation of fetal tissues is important because m-hPSCs tend to dominate cultures quickly [227,228].

### 3.2. Suggested Modes of Action of hPSCs-Based Therapies

Accumulated data support the observation that most stem and stromal cells therapies, leading to the repair of compromised tissues and organs, are not necessarily associated with their physical integration in the repaired organs or with replacing cells in the damaged tissues, as some studies seem to propose [188,229,230,231,232]. Rather than their pro-regenerative activity and high potency being used to support the healing of damaged tissues, their angiogenic and anti-inflammatory effects seem to be associated with their ability to respond to stress signal and to subsequently induce pro-regeneration and anti-inflammatory effects by releasing the corresponding growth factors, cytokines, and extracellular vesicles [114,233].

### 3.3. Pro-Regenerative Therapy with hPSCs from Commercial Sources 

Many studies used commercially produced hPSCs, such as the PLX cells produced by Pluristem or PDA-1 cells by Celgene. Commercial companies purposely do not fully disclose in peer-reviewed scientific publications the exact methods of isolation, quality measures, and production of such cells. However, production methods employed will have an impact on cell characteristics. This tendency of corporate entities to disclose as little as possible critical information on the cells is a major drawback for the comprehension of their activity and exact composition, which may be crucial for understanding their mode of action. In general, the f-hPSCs provide systemic effects when applied in highly vascularized tissues [114,206]. Nevertheless, this may not be always the case. For instance, IM injection of f-hPSCs failed to reduce the cerebral damage in a mouse model of induced brain inflammation, probably due to the blood–brain barrier. In contrast, the efficacy of f-hPSC was highly significant after intracerebral injections [234].

### 3.4. hPSCs Therapy as an Example of Their Use for Mitigation of Acute Radiation Syndrome (ARS)

BM regeneration in ARS has been a therapeutic challenge. Since hematopoietic stem cells are dependent of the support of BM stromal cells, the administration of bmMSCs was proposed to help recover severe radiation-induced damage [235,236]. Intramuscular (IM) injection of f-hPSCs mitigated experimental ARS by induction of regeneration of the radiation-depleted BM [206,237,238]. The xenogeneic therapy was based on expanded hPSCs, enriched with high proportion of f-hPSCs, termed PLX-RAD (or commercially tagged as PLX-R18) [206]. The injected cells remained at the injection site without migration to other organs [239,240] (Figure 4). The product with a high proportion of f-hPSCs out-performed significantly the benefit of the product containing m-hPSCs only [206,237,238]. The effect seemed to relate to the response of the hPSCs to stress signals induced by the experimental injury, which provoked the release of the regenerative factors described above. The kinetics of the hPSC responses correlated well with the stress condition of irradiated mice. Of note, when injected to non-irradiated mice, the secretome of the f-hPSCs was not activated [114].

Based on these results, Gorodetsky’s group disclosed a simple procedure for the direct isolation and expansion of pure population of f-hPSCs, which contained only cells from the chorionic plate [116]. The intramuscularly injected pure f-hPSCs induced the full regeneration of the hematopoietic system and BM in an ARS mouse model after lethal irradiation (Figure 4D–F) [130,206]. The use of f-hPSCs isolated from Wharton Jelly was also proposed for the mitigation of radiation damage of lower doses, which seems to confirm a higher pro-regenerative effect of f-hPSCs derived from placenta [199,217]. Pre-clinical studies with pure f-hPSCs seem to yield beneficial effects, surpassing the effects of injections of G-CSF only [241]. 

### 3.5. The Anti-Inflammatory Effect of the hPSCs

Various types of MSCs and PSC have been claimed to have anti-inflammatory effects. Most of these studies related to inflammatory lung failure [53,73,242,243,244,245]. These treatments have been tested successfully also for other severe inflammatory conditions, including colitis and skeletal inflammation [53,243,246,247,248,249,250]. In induced CNS inflammation, a significant effect of the cells was reported only upon direct intra-cerebral injections. It can be concluded that upon IM injection of f-hPSC either the penetration of the cells’ secretome to the brain or the relay of stress signals from the brain to activate the cells in remote injection sites (i.e., muscle) are blocked by the blood–brain barrier (BBB) [233]. To overcome the possible BBB obstacle and to increase the effect of the therapeutic cells, their injection into the olfactory bulb was proposed [251]. 

Accumulating evidence suggests that the diseased tissue targeted by MSCs or the current health condition of the patient treated—in pre-clinical situations this would be the stage of a disease/defect induced in the experimental model—may modulate the regenerative potential of such cells. MSCs applied during flare of inflammation, in different stages of an (autoimmune) disease seem to act differently from MSCs applied during remission or injected in a healthy environment [252].

Secondary inflammatory processes may derive from various disorders such as diabetes mellitus. The hPSCs were shown to be useful in treating the inflammatory complications of diabetes [56,74,253,254,255]. An extensive study showed effects of f-hPSCs on pancreas regeneration and experimental diabetes in a rat model [256].

Typical to placental stromal cells, the f-hPSCs were found to express mesenchymal stromal cell markers [205], as well as CD166, a common marker of mesodermal cell types and MSCs [115,257]. They also are positive to CD146 (MCAM), expressed in different mesenchymal cells, including pericytes. The negligible expression of HLA-G (class I) and HLA-DR in f-hPSCs seems to contribute to the immune-tolerance of these cells in both allogenic and xenogeneic injections [258,259]. Based on these insights, the f-hPSCs that were isolated only from the fetal placental tissues and further expanded can be proposed as an optimal indirect pro-regenerative and anti-inflammatory treatment for various conditions [116,130,234]. Other reports on the effect of these cells on the recovery from inflammatory bowel disease and reduction of skin and hair follicle damage following high dose local irradiation are now prepared for publication. Preclinical studies comparing the efficacy of atMSCs vs PSCs to regenerate the sphincter muscle in an animal model for urinary incontinence are well under way. Interestingly, MSCs derived from the amniotic membrane produced a secretome suitable for regeneration of musculoskeletal tissues [260].

The biological rational for the expectation of better responses from f-hPSCs is that these cells, which originate of the chorionic plate, share and connect to the fetus’ circulation. This may explain the apparent higher activity of f-hPSCs, which can respond to the fetal stress signals carried in the fetal circulation along the development of the pregnancy, in confronting various acute stress messages, such as depleted function of the hematopoietic system [261,262,263]. This may explain why preparations enriched for f-hPSCs were more effective than preparations of m-hPSCs consisting of cells from maternal tissues only [206].

## 4. Summary and Conclusions

The mainstream approach of attempting to use stem cells implants for the reconstruction of degenerated tissue and for the replacement of cells in failing organs has so far yielded disappointing results. This diverted the focus of cell therapy from dealing with the differentiation potential (“stemness”) of the cells while attempting to develop treatments based on the indirect and paracrine regenerative effect of this cell therapy. For this goal, MSCs from different sources seem to pose as preferred candidates based on their role in wound healing and tissue regeneration in many disorders. The most studied cells are the bmMSCs, isolated from BM in low numbers and expanded to high numbers by passaging. A wide range of projects demonstrated that MSCs isolated from various other tissues may have pro-regenerative, anti-inflammatory, and immune-modulating activity that surpasses the effects observed initially with bmMSCs.

The mature, full-term fresh placenta is an easily available disposable healthy bulk organ. It contains cells from comparably young donors. Such cells are sensitive to different stimuli and seem to adapt their secretome to the regenerative need of the donor. The f-hPSCs seem to be the most potent candidates for PSCs-based therapies. The rational for this difference may lie in the role of the fetal part of the placenta in the direct support and nourishment of the developing fetus. Another advantage of f-hPSCs is their lower rejection rates, since they derive from the fetal placenta, which is in a very tight interaction with the allogeneic maternal tissue and blood that would otherwise reject a foreign tissue. This renders the f-hPSCs an ideal source of allogeneic cells for (pre-) clinical studies.

In continuation of earlier studies on the use of MSC for regenerative medicine, m-hPSCs and f-hPSCs have demonstrated very impressive activities in treating severe complications associated with tissue degeneration and severe inflammation. The mechanistic explanation of their activity and their diverse applications deserve further thorough investigation.

## Figures and Tables

**Figure 1 ijms-22-05302-f001:**
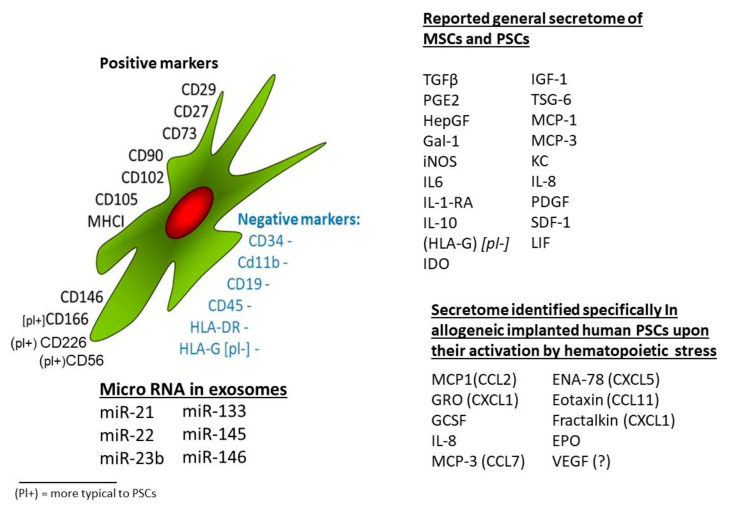
**A shared phenotype of MSCs surface markers and secretome from different cell sources.** MSCs from different sources seem to have a wide range of shared features in their cell surface markers, secretion profile of growth factors, cytokines, and miRs. The significant difference between the MSCs from different tissue sources may lie predominantly in the levels and profile of the factors secreted and the degree of their activation by the induction of these factors secretion. The data presented were assembled mainly based on the reviews on MSCs of Pittenger et al. [87], Eleuteri and Fierabraci [88], and on the studies on PSCs of Pinzur et al. [114] and Adani et al. [116].

**Figure 2 ijms-22-05302-f002:**
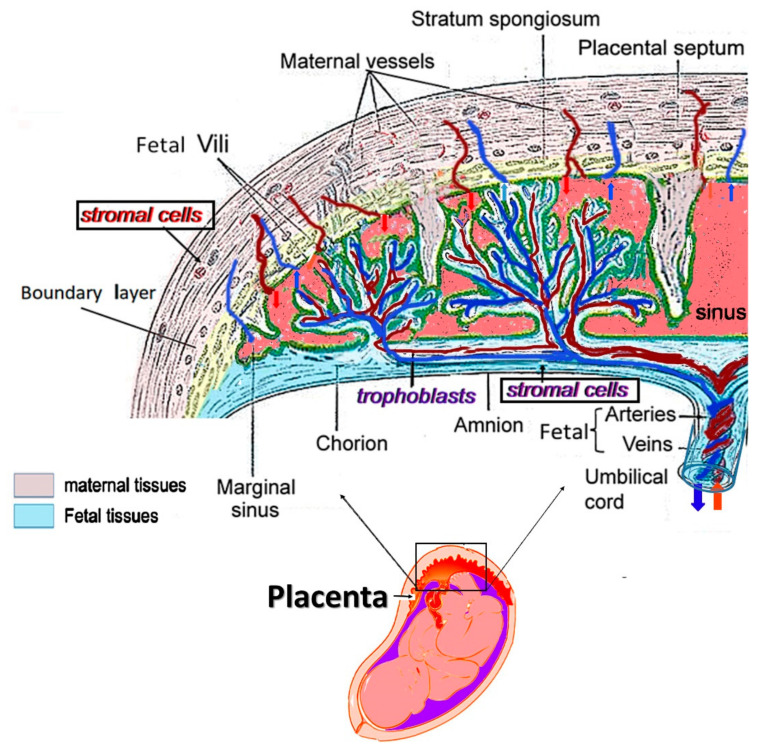
**The structure** **of full-term placenta and the tissues from where hPSCs are isolated.** The structure of the placenta. The placenta is composed of the fetal and maternal tissues interconnected with a border and separate network of blood vessels. It is clearly demonstrated that the source of the stromal cells, in case the tissue samples taken are not carefully dissected out, may contain the combination of the fetal and maternal cells from two separate individuals.

**Figure 3 ijms-22-05302-f003:**
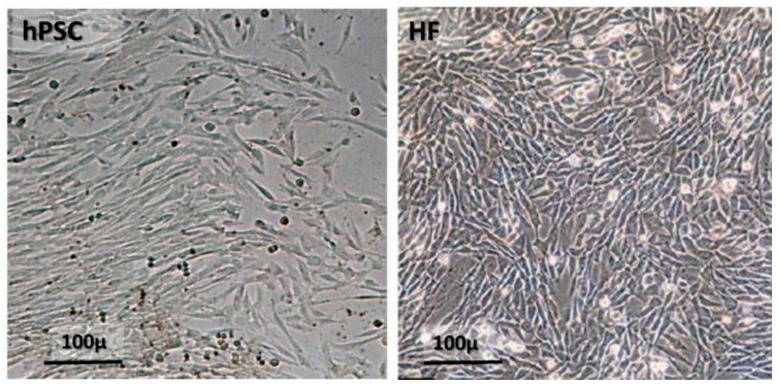
**Difference of shape of cultured PSCs vs bmMSCs.** The difference in the shape of freshly cultured isolated human hPSCs and human skin fibroblast (HF). The hPSCs spread in culture can reach a size of up to ~100 µm, while the fibroblasts are much smaller in size and grow in much more condensed cultures. Nevertheless, both cell types show almost similar surface markers, except for CD166, which is expressed more in PSCs (see also Figure 1).

**Figure 4 ijms-22-05302-f004:**
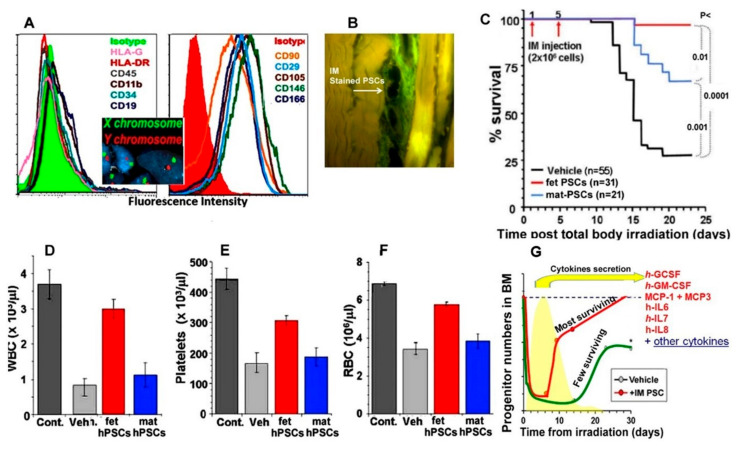
**An example of successful hPSCs-based treatment for mitigation of acute radiation syndrome by remote PSCs injection.** Example of the indirect systemic effect of the f-hPSCs as demonstrated in studies on the mitigation of acute radiation syndrome. Mice were total body irradiated with a lethal dose and then treated by fetal human PSCs. The X/Y chromosomes staining and cell surface markers profile of the cells are presented in (**A**). The CFSE-stained f-hPSCs seemed to stay in the injection site in the muscle, from where their secretome must have reached the circulation to help regenerate the BM (**B**). The mice treated with f-hPSCs had better mitigation of ARS than PSCs isolated from the maternal placenta (**C**), with impressive recovery of white blood cells, platelets, and red blood cells (**D**–**F**). (**G**) The cytokine production (shown as yellow peak) represents the kinetics of major related secretome in the plasma. The peak of the kinetics of plasma levels of the f-hPSCs secreted human cytokines coincided with the apparent recovery of the cell-treated pre-irradiated mice and the increase of their BM progenitors (red line) in comparison with the progenitor number in the few surviving non-treated pre-irradiated mice (green line). The figure presented was re-plotted based on the data in the reports of Gaberman et al. [206], Pinzur et al. [114], and Adani et al. [116]. * indicates a significant difference between cell-treated (+IM PSC) and sham- treated (vehicle) animals.

## Data Availability

Not applicable.

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
