# Peer review of "Allogenic Use of Human Placenta-Derived Stromal Cells as a Highly Active Subtype of Mesenchymal Stromal Cells for Cell-Based Therapies"

_ijms, 2021, doi:10.3390/ijms22105302_

Round 1

Reviewer 1 Report

Revision IJMS

In the manuscript titled “Allogenic Use of Human Placenta-derived Stromal Cells as a Highly Active Subtype of Mesenchymal Stromal Cells for Cell Based Therapies”, the authors review the potential application of allogeneic placental derived MSC in clinic.

The article is of interest for this journal, however there are some gaps that must be revised

Minor revisions are required

-In general, the authors discuss and compare MSCs, particularly BM-MSCs with placenta-derived MSCs. This comparison is very appreciable, but at the same time generates a whole series of ambiguities when considering that perinatal MSCs, even those limited to placental tissue derivation, represent a "world" unto themselves when considering the number and type of cells that are isolated ((10.3389/fbioe.2020.610544)). In addition, the growing interest in these cells is evidenced by the increased number of published clinical trials and scientific studies. This in itself would require a revision of the article at least for the part pertaining to cells isolated from the placenta

In addition, the authors should also consider an important point represented by the requirement of BM-MSCs for priming to exert an active functional effect ( 10.1038/ni.3002 ) compared with what is observed for example for hAMSCs that do not require this step.                                                                                    

Therefore, the authors should better explain the differences between the various types of mesenchymal cells isolated from the placenta, which also have peculiar properties, especially from the perspective of immunomodulation.

-The authors should use the acronym hPC to refer to cells isolated from human term placenta instead of hPSCs as reported in the latest consensus paper published by Silini et al (10.3389/fbioe.2020.610544). In addition, when referring to placenta-derived MSCs the authors should use the acronym hPMSC also from Silini et al (10.3389/fbioe.2020.610544).

-At line 225, the authors stated "It can be hypothesized that PSC derived microvesicles also contribute to angiogenesis." A paper was recently published that characterizes the secretome of hAMSCs and highlights that both EVs and the free fraction of EVs contain factors responsible for their immunomodulatory activity (10.1002/sctm.20-0390). The authors need to discuss this point as well.

-Figure 2. The figure used appears to be derived from a book. Do the authors have any reference that needs to be stated or does the author have the copyright? Either way, for both figures, the authors need to improve the overall quality of the images used.

Author Response

Reviewer 1:

Revision IJMS

In the manuscript titled “Allogenic Use of Human Placenta-derived Stromal Cells as a Highly Active Subtype of Mesenchymal Stromal Cells for Cell Based Therapies”, the authors review the potential application of allogeneic placental derived MSC in clinic.

The article is of interest for this journal, however there are some gaps that must be revised

Minor revisions are required

-In general, the authors discuss and compare MSCs, particularly BM-MSCs with placenta-derived MSCs. This comparison is very appreciable, but at the same time generates a whole series of ambiguities when considering that perinatal MSCs, even those limited to placental tissue derivation, represent a "world" unto themselves when considering the number and type of cells that are isolated ((10.3389/fbioe.2020.610544)). In addition, the growing interest in these cells is evidenced by the increased number of published clinical trials and scientific studies. This in itself would require a revision of the article at least for the part pertaining to cells isolated from the placenta.

Reply: We thank reviewer 1 for the detailed suggestions to improve the original version of our review. As outlined in table 1 of the paper published by Silini et al. in 2020 (Frontiers in Bioening Biotechnol 8: 610544), many different cells are found in placenta tissue. We therefore refer to the work by Paola Parolini’s group and added a corresponding reference in the revised manuscript in chapter 3.1 (see new reference #187 ). However, the aim of this study is not to give a detailed report on the variety of different types of cells described in placenta vs. bone marrow, adipose or other tissues. As the reviewer states, this topic was covered by a conclusive review by Silini and colleagues last year. The aim of our review to concentrate of the placenta-derived stromal cells, called hPSCs, in the context of wound healing, immune modulation and tissue regeneration, with reference to other mesenchymal stromal cells. A larger in-depth review of the large variety of mesenchymal stromal cells may be of great interest, but is may deviate from the focus of the main topic of this manuscript.

In addition, the authors should also consider an important point represented by the requirement of BM-MSCs for priming to exert an active functional effect ( 10.1038/ni.3002 ) compared with what is observed for example for hAMSCs that do not require this step. 

Reply: We greatly appreciate this comment of the reviewer. The study by Wang et al. (Nature Immunol. 15:1009; 2014) describes the potential of bmMSCs in modulation of inflammation and the plasticity of MSCs in individual responses to a given clinical situation. This aspect has not been discussed in detail in the original version of our manuscript. This very important point indeed needs to be addressed, when considering the specific cell types of bmMSCs as active components of a therapy. To address in more details. We now refer to this publication in chapter 3.6 of the revised manuscript and added a new related reference (ref 197).

Therefore, the authors should better explain the differences between the various types of mesenchymal cells isolated from the placenta, which also have peculiar properties, especially from the perspective of immunomodulation.

Reply: This point refers to an issue, which was already addressed above. This is indeed an important issue in the research of placenta stromal cells. And above all, an aspect that for sure merits in-depth studies and detailed publications. However, this review already exceeds the quota of number of words allowed by more than 500 words with about 260 references. To keep it relatively short and attractive to the average, even not specialized, readers who may have interest in the main aspects of placenta-derived cells investigation was the focus, Unfortunately, we also had to limit ourselves to a selection of topics, focusing more on (pre-)clinical applications of these cells, including the EMA and FDA approved studies.  

-The authors should use the acronym hPC to refer to cells isolated from human term placenta instead of hPSCs as reported in the latest consensus paper published by Silini et al (10.3389/fbioe.2020.610544). In addition, when referring to placenta-derived MSCs the authors should use the acronym hPMSC also from Silini et al (10.3389/fbioe.2020.610544).

Reply. The nomenclature is for sure a very important point to discriminate cells isolated from different tissues and produced under different conditions from other cells. hPC is an acronym of human placenta cells. This does not refer to the phenotype of the placental stromal cells. Since the placenta is composed of many cell types including endothelial cells, trophoblasts and we refer to only the stromal cells we decided, in all our cited works in this review to use h-PSC, which include the abbreviation “stromal” to better define the cells. To remain in line with our previous work, which are cited by Gorodetsky’s group, we prefer to stick to the terminology in review that describes these cells. For those who are used to the term hPC, we introduced a reference to this term in the description of the cells and referred to it as an alternative terminology of these cells.

-At line 225, the authors stated "It can be hypothesized that PSC derived microvesicles also contribute to angiogenesis." A paper was recently published that characterizes the secretome of hAMSCs and highlights that both EVs and the free fraction of EVs contain factors responsible for their immunomodulatory activity (10.1002/sctm.20-0390). The authors need to discuss this point as well.

Reply: The secretome and clinical efficacy of amnion membrane-derived MSCs is not the focus of our review. However, the aspect is very interesting corroborating that mesenchymal stromal cells from different sources may induce very different pro-regenerative effects. Therefore, we added this aspect to the revised manuscript in section 3.6.

-Figure 2. The figure used appears to be derived from a book. Do the authors have any reference that needs to be stated or does the author have the copyright? Either way, for both figures, the authors need to improve the overall quality of the images used.

The figure addressed may have contained certain parts which have been adopted from previous publications. Figure 2 is now based on a new original drawing. Figure 4 contains some graphics adopted from a publication in PlosONE which transferred the rights to the author (Gorodetsky). A reference to this publication as the source of the data in this figure is given.

Reviewer 2 Report

Gorodetsky and Aicher present a review on the characterization, mechanism of action and possible terapeutic uses of placental stromal cells. The paper is a well-writen, comprehensive review of the literature that can be of interest to both stem cell researchers and clinicians.

Besides minor typos, some modifications on the organization of the introduction would help the author’s points to come across more clearly.

Please see below my specific comments.

Major

1- In general, the introduction seemed to focus on the current pitfalls of stem cell therapies, including for tissue engineering or tissue repair. Mentioning other uses of stem cells in the introduction, including the more successful paracrine applications, would better set the tone to the rest of the review. This have been done, in part, in section 1.2. Is it possible to merge sections 1.1 and 1.2 into a single introduction?

2- Similarly, it would be important to briefly mention the advantage of local applications in the introduction (as will be mentioned in section 3.6) in comparison to systemic (lines 47-49).

3- A mention to differentiation of stem cells into the same lineage in cell therapies (such as BMSC in bone implants, ref [63]) instead of trans-differentiation was missing from the review.

Minor

1- “Almost unlimited numbers” (line 18) could be described more precisely, such as “abundantly available”.

2- Was reference [1] attributed correctly?

3- A few typos were found: in-vitro instead of in vitro (line 36), Figure1 instead of Figure 1 (line 132), lecvel (line 136), µ instead of µm (line 311 and Figure 3).

4- The subtitle 2.3 indicates that the surface markers and secretome are going to be addressed, but surface markers are characterized in 2.3 and the secretome in 2.4.

5- In Figure 1 “positive markers” should be underlined, and the use of upper/lowercase could be standardized in Figure2.

6- Figure 4 would benefit from a more descriptive legend, which seems to have been added to the body of the text of the section 3.5 (paragraph starting on line 373).

Author Response

Reviewer 2:

Comments and Suggestions for Authors

Gorodetsky and Aicher present a review on the characterization, mechanism of action and possible terapeutic uses of placental stromal cells. The paper is a well-writen, comprehensive review of the literature that can be of interest to both stem cell researchers and clinicians.

Besides minor typos, some modifications on the organization of the introduction would help the author’s points to come across more clearly.

Please see below my specific comments.

Major

1- In general, the introduction seemed to focus on the current pitfalls of stem cell therapies, including for tissue engineering or tissue repair. Mentioning other uses of stem cells in the introduction, including the more successful paracrine applications, would better set the tone to the rest of the review. This have been done, in part, in section 1.2. Is it possible to merge sections 1.1 and 1.2 into a single introduction?

Reply: We appreciate the comments of reviewer 2 and revised the manuscript accordingly: In the second sentence (i.e., lines 13,14) we exactly described in the original version of the abstract what this reviewer suggested: the paracrine action of such cells. Unfortunately, we used other words. We therefore revised the manuscript to include the term “paracrine” to the abstract of the revised version. Moreover, section 1.1. and 1.2 were fused in the revised review as suggested.

2- Similarly, it would be important to briefly mention the advantage of local applications in the introduction (as will be mentioned in section 3.6) in comparison to systemic (lines 47-49).

Reply: We thank the reviewer 2 for this suggestion. In the context of hematopoietic stem cells, local administration is not used at all. But for non-HSCs local administration is the ticket. We revised the manuscript accordingly.

3- A mention to differentiation of stem cells into the same lineage in cell therapies (such as BMSC in bone implants, ref [63]) instead of trans-differentiation was missing from the review.

Reply: Reviewer 2 raises here a very important question that deals with the “stem cell nature” and the basic concepts of “stemness” and differentiation potentials of MSCs. We therefor appreciate this comment a lot. But the authors are not convinced that under rather physiological conditions MSCs, produced under standard conditions (see ref. 64 : consensus paper of the ISCT in 2006 by Massimo Dominici and colleagues), will actually TRANS-differentiate to cells of other germ lines. According to the current literature we use the term trans-differentiation when for instance a mesenchymal cell becomes an ectodermal cell or an endodermal cell. This happens in cancer but under strict measures not with MSCs: Expression of e.g., nerve cell growth factor receptor (alias CD271, excellent work e.g. by Hans Bühring’s group) on subsets of bmMSCs or MSCs from other sources does not make such cells functional nerve cells. MSCs are clearly cells of the mesodermal lineage. In vitro they may generate cell types such as osteoblasts, chondrocytes, adipocytes or even smooth muscle cells. But MSC do not differentiate along all of these differentiation pathways with really robust efficacy, nor with a stable phenotype. For instance, upon in vitro differentiation along the chondrogenic lineage, such cells tend to become terminally differentiated fibroblast-like cells expressing very little type II collagen or aggrecan – important factors of a cartilage cells. Comparably, when differentiating MSCs along the smooth muscle cell lineage, cells can be generated that express some of the proteins needed for contraction, some of the ion channels needed for communication, and thy even contract to a certain degree. But when compared to a real smooth muscle cell isolated from smooth muscle tissue, significant differences remain (Brun, PlosOne doi; 10.1371/journal.pone.0145153). We kind of agree to the work published by Muraglia (J. Cell. Sci. 113:1161; 2000) or Bianco (Nat. Medicine 19:35; 2013) and others which showed that the most efficient differentiation of MSCs yields osteoblast. This can be observed in mice in vitro and by consecutive transplantation experiments in vivo (see: Bianco et al. 2013). Other differentiations are possible in vitro, but seem much less favored. For real trans-differentiation of MSCs powerful genetic techniques such as transfections by differentiation promoting factors, iPSC technology…) are required. But such applications are not in the focus of this review.

Minor

  • “Almost unlimited numbers” (line 18) could be described more precisely, such as “abundantly available”.

Reply: The wording suggested by reviewer 2 changes the flavor of the abstract to a more realistic description of the yield of cells produced from an individual placenta. We therefore revised these lines accordingly.

  • Was reference [1] attributed correctly?

Reply: the reference was checked by the authors and seems ok.

3- A few typos were found: in-vitro instead of in vitro (line 36), Figure1 instead of Figure 1 (line 132), lecvel (line 136), µ instead of µm (line 311 and Figure 3).  

Reply: we really appreciate these suggestions and fixed all typos listed

  • The subtitle 2.3 indicates that the surface markers and secretome are going to be addressed, but surface markers are characterized in 2.3 and the secretome in 2.4.

Reply: This is an excellent point of critique which the authors obviously missed. We of course revised the subtitle accordingly

  • In Figure 1 “positive markers” should be underlined, and the use of upper/lowercase could be standardized in Figure2.

Reply: Annotations in artwork can be designed in different ways to guide the reader through the content with ease. We decided to present in Fig. 1 positive markers by black letters and neg. markers by blue ones. We prefer to keep this color coding and not to increase the spacing of the makers by underlining, as we list many factors on the left side of the cell. Underlining would possibly kind of “push the miRNAs and exosomes out of the picture”

  • Figure 4 would benefit from a more descriptive legend, which seems to have been added to the body of the text of the section 3.5 (paragraph starting on line 373).

Reply: We revised the legend to Fig. 4. As suggested.
